# Microstructure Dependence of Output Performance in Flexible PVDF Piezoelectric Nanogenerators

**DOI:** 10.3390/polym13193252

**Published:** 2021-09-24

**Authors:** Yijing Jiang, Yongju Deng, Hongyan Qi

**Affiliations:** 1Institute for Functional Materials, School of Physics and Mechanical and Electronical Engineering, Hubei University of Education, Wuhan 430205, China; 201822111310179@stu.hubu.edu.cn (Y.J.); qihongyan@hue.edu.cn (H.Q.); 2School of Materials Science and Engineering, Hubei University, Wuhan 430062, China

**Keywords:** flexible piezoelectric energy harvesting, flexible, PVDF, microstructure

## Abstract

Flexible piezoelectric nanogenerators have attracted great attention due to their ability to convert ambient mechanical energy into electrical energy for low-power wearable electronic devices. Controlling the microstructure of the flexible piezoelectric materials is a potential strategy to enhance the electrical outputs of the piezoelectric nanogenerator. Three types of flexible polyvinylidene fluoride (PVDF) piezoelectric nanogenerator were fabricated based on well-aligned nanofibers, random oriented nanofibers and thick films. The electrical output performance of PVDF nanogenerators is systematically investigated by the influence of microstructures. The aligned nanofiber arrays exhibit highly consistent orientation, uniform diameter, and a smooth surface, which possesses the highest fraction of the polar crystalline β phase compared with the random-oriented nanofibers and thick films. The highly aligned structure and the large fraction of the polar β phase enhanced the output performance of the well-aligned nanofiber nanogenerator. The highest output voltage of 14 V and a short-circuit current of 1.22 µA were achieved under tapping mode of 10 N at 2.5 Hz, showing the potential application in flexible electronic devices. These new results shed some light on the design of the flexible piezoelectric polymer-based nanogenerators.

## 1. Introduction

Confronting the intemperate consumption of fossil energy and pollutants generated from their use [1], together with the rapid development of flexible and wearable electronics [2] and artificial intelligence, there is an urgent requirement to replace fossil fuels [3]. Until now, various new sources of energy have been successfully demonstrated, including solar, wind, geothermal, tidal, nuclear, and mechanical energies [4]. The most widely distributed mechanical energy can be harvested from the environment and biomechanical movement, so it is an attractive alternative for self-powered portable/wearable personal electronics [5]. Converting ambient mechanical energy into useful electricity energy by using piezoelectric materials has become a research focus due to its high performance, lightweight, and sustainable high energy conversion efficiency [6].

Compared with inorganic piezoelectric materials, organic piezoelectric materials, such as poly(vinylidene fluoride) (PVDF) and its copolymers, are preferable for wearable nanogenerators because of their naturally flexibility and biocompatibility, chemical stability, and easy synthesis [4,7]. They also exhibit high purity, lightweight and resistance to chemical solvents and stability under high electric fields [8]. More importantly, availability of PVDF is sufficient for mass applications with a relatively low price [9]. All these advantages are the main reasons for its more extensive application compared to that of other copolymers. PVDF composed of the [–CH_2_–CF_2_–] monomers is a semicrystalline polymer, while the crystalline PVDF usually exhibits four different phases, i.e., alpha (α), beta (β), gamma (γ), and delta (δ) phases, depending on the chain conformation during crystallization [10]. The common α phase is the most thermodynamically stable phase, while the β phase containing oriented fluoride and hydrogen unit cells aligned along the carbon backbone is an electrically active polar phase with a strong dipole moment [11]. Furthermore, enormous electrical properties such as pyro-, piezo-, and ferroelectric properties are found in the β phase of PVDF and these properties have been widely investigated and extensively utilized in actuators, sensors, energy harvesters and transducers [11]. Therefore, the β phase content in the PVDF is of primary importance for piezoelectric-based applications. In this case, a wide range of additional processes, including high electric field poling [12], mechanical stretching along one- or two-directions [13], epitaxy process [14], electrospinning [9] and doping with organic and inorganic materials [15,16]. Comparing these processing methods, electrospinning is considered to be one of the easiest and simplest techniques used to increase the β phase content [17,18]. In the electrospinning process, the applied high electric field stretches the fluid to form the nanofiber structure. The electrospinning causes natural poling which is helpful for the transforms from non-polar phases into the polar β phase.

Researchers are focusing on the improvement of the electrical outputs of the flexible nanogenerators by optimizing the piezoelectric materials both by designing new flexible piezoelectric materials and forming inorganic-organic composites, designing 3D electrodes, etc. As summarized in the review papers [2,4,5,7], the reported nanogenerators usually have the features of high open open-circuit voltage (from millivolts to 200 volts) but low short-circuit current and low power density. All efforts have been put into improving the conversion ability of the flexible piezoelectric materials. Besides increasing the content fraction of the β phase, various strategies have been used to enhance the piezoelectric output performance of the PVDF-based flexible nanogenerator, such as designing high piezo-response flexible nanocomposites and controlling the microscopic structure of the piezoelectric materials to fabricate the flexible nanogenerators [4]. Manipulation of the microstructure can not only control the connection of the piezoelectric materials but also increase the deformation of the piezoelectric membranes when it is subjected to an external mechanical stress. Following this conception, hierarchically architected microstructures [19], highly aligned nanofibers with micropatterned three-dimensional interdigital electrodes [20], and nanofiber film/well-ordered cylindrical cavity structure [21] have already been designed and relatively high output performances have been achieved.

Electrospinning is an efficient way of controlling the microstructure of the nanofibers by tuning the characteristics of the polymer solution, the electrospinning parameters and also the ambient condition. Especially for the aligned polymetric nanofibers, several additional techniques have been employed to fabricate aligned nanofibers such as magnetic, mechanical and electrostatic ways during electrospinning [22,23,24]. Furthermore, the ferroelectric polymer nanofibers such as PVDF and its copolymers can be poled simultaneously in the electrospinning process without additional poling process, and thus enhance the piezoelectric response. Although the highly aligned piezoelectric nanofibers have been proved to show high output performances [20,21], a comparative study of the microstructure effects on the electrical outputs of the PVDF-based nanogenerator is still lacking.

In this article, the microstructures of the PVDF nanofiber mats have been controlled by tuning the nanofiber collecting parameters to form the highly aligned nanofibers, random oriented nanofibers and thick films. Accordingly, three types of PVDF-based nanofiber mat of different microstructures have been realized, and the output electrical performances have been compared between nanogenerators based on the highly aligned nanofibers, random oriented nanofibers and thick films. It is shown that the well-aligned nanofiber nanogenerators present the highest output voltage of 14 V, a short-circuit current of 1.22 μA, compared with the nanogenerators fabricated by random oriented nanofibers and thick films. The well-aligned nanofiber nanogenerators also exhibit superior stability and durability. Our research not only provides a valuable, reliable and repeatable but simple and low-cost method without any secondary processes to control the microstructure of the piezoelectric nanofiber of high mechanical energy converting ability, but also sheds some light on the design of the flexible piezoelectric polymer-based nanogenerators and portable/wearable personal electronics.

## 2. Materials and Methods

The PVDF nanofibers were synthesized by the advanced electrospinning technique. In the electrospinning process, a high voltage was applied to the polymer solution and the liquid droplets were stretched by the electrical field. At a critical point a stream of liquid erupts from the surface to form a Taylor cone. With the increase of the electric field to a critical point, the surface tension is not large enough to maintain the Taylor cone and the polymer jet is initiated and drawn from the nozzle tip to the collector. While traveling to the collector, the solution jet solidified due to the fast evaporation of the solvent and was deposited on a collector [25].

PVDF (Kynar Flex 2801) was purchased from Arkema Inc. (Colombes, France). Dimethylformamide (DMF) was purchased from Sigma-Aldrich (Burlington, MA, USA). Acetone was acquired from Sinopharm Chemical Reagent Co., Ltd. (Shanghai, China). DMF together with acetone were used as a solvent to dissolve the PVDF powders. A PVDF solution was prepared by dissolving 1 g PVDF powder into a mixture of 3.5 mL of DMF and 1.5 mL of acetone and then stirring until a homogeneous solution was obtained at room temperature. The solution was transferred into a 5 mL syringe for electrospinning. An electrospinning machine (TL-Pro, Tongli, Shenzhen, China) was used to fabricate the PVDF nanofibers. In the electrospinning process, the air humidity was adjusted to around 40%, the temperature was adjusted to around 30 °C. A nozzle with a nominal inner diameter of 0.4 mm was used and the injection speed was maintained at 0.3 mL/h. A high-voltage generator and a syringe pump were used to control the electrospinning rate. The voltage applied between the syringe needle and the collector was about 15 kV; a distance of 10 cm was maintained between the needle and the collector. The rotation speed of the collector was used to control the alignment of the nanofiber array. For the random nanofibers (r-PVDF), the rotating speed was set to 0, while it was set in the range of 1000~2500 rpm for the aligned nanofiber array. The highly aligned nanofibers with a rotating speed of 1700 rpm were short for a-PVDF. The sample information has been summarized in Table 1. The as-prepared PVDF nanofibers thickness was in a range of 45~50 µm, which was dependent on the size of the electrode and spinning time. The electrospun membranes were desiccated at 60 °C for 24 h. To prepare the PVDF films (f-PVDF), the as-prepared solution was dropped into the Petri dish, which was then sealed and heated up to 30 °C for 24 h. The dry films were peeled off from the Petri dish and attached with Pt-coated PI substrate on both sides to form the thick-film nanogenerator.

The nanogenerators were fabricated based on the dry electrospun membranes with 2.5 cm × 2.5 cm in size. Two copper wires were attached to each side of the film while an aluminum foil electrode with ~50 µm thickness was attached to each side of the film. Then, the membrane consisting of two electrodes and connecting wires was packaged in polyimide (PI) tape. The preparation of the PVDF nanofibers and the fabrication of the nanogenerator are summarized in Figure 1.

X-ray diffraction (XRD, Bruker D8 Advance diffractometer, CuKα radiation, Carl, Germany) and Fourier transform infrared (FTIR) spectroscopy (Nicolet iS50, Waltham, MA, USA) were used to characterize the phase structure of the PVDF materials. Field-emission scanning electron microscopy (FESEM, Sigma 500, Carl, Germany) was used to observe the morphology of the PVDF membranes. The Raman spectra of PVDF membranes were recorded using a Jobin Yvon LabRAM HR800 Raman spectrometer. The piezoelectric output of the nanogenerators was measured on a home-built platform in a repetitive tapping mode of 10 N at 2.5 Hz. A digital force gauge was used to measure the dynamic pressing force applied to the a-PVDF (Aipu Metrology Instrument Co., Ltd., Hangzhou, China). The open-circuit voltages of the nanogenerators were measured with a digital storage oscilloscope (TBS1072B, Tektronix, Beaverton, OR, USA). The short-circuit current was measured by an electrometer (2450, Keithley, Beaverton, OR, USA).

## 3. Results

Figure 2 shows SEM images of the PVDF nanofibers and a cross-sectional image of the PVDF film. Images in Figure 2a–e are the nanofibers obtained for collector speeds of 0, 1000, 1500, 1700 and 2500 rpm, respectively. By compiling the statistics from over 120 nanofibers in a set of SEM images for each sample prepared at various collector rotation speeds, the diameters of the nanofibers were found to be in the range from 50 nm to 250 nm with a smooth surface. Preferential alignment in a particular direction was realized with high collector rotation speed. The nanofibers were parallel to each other and arranged in a group, and this characteristic became increasingly obvious with the increase of collector speed. The degree of alignment for the nanofiber arrays was quantitatively analyzed by the statistic of the distribution angle, as shown in the insets of each image. The most highly aligned nanofiber array was observed in a membrane with 1700 rpm rotation speed. The angle between most nanofibers and the axial direction was mainly concentrated in the range of −20° to 20°. The following crystal structure and electrical output performance of the well-aligned PVDF nanofibers are based on the nanofibers collected with a rotation speed of 1700 rpm, which is as short as a-PVDF. Figure 2f highlights the microstructure of the film cross-section based on SEM, which shows the continuous and dense structure for the f-PVDF film. 

The crystal structure of a-PVDF nanofibers, r-PVDF nanofibers, and f-PVDF thin film were characterized by using XRD, as shown in Figure 3a. The diffraction peaks of the (110), and (200) planes for the pure PVDF are detected around the diffracted angles of 2*θ*~19.9°, and ~20.6°, respectively. These peaks confirm that the α-crystalline and β-crystalline phases exist in pure PVDF [10]. The diffraction peaks shown in Figure 3a can be indexed to the α-phase and β-phase of PVDF, and no secondary phase was detected. The characteristic peak of β phase of a-PVDF nanofibers at 20.6° is distinct while the peak for α phase of r-PVDF nanofibers, and f-PVDF thin film at 19.9° is prominent in the patterns, revealing that the a-PVDF nanofibers possess the highest diffraction intensity of β phase comparing with that of the r-PVDF nanofibers and thin films [26,27,28]. 

To further confirm the information of the crystallographic structure of the a-PVDF nanofibers, r-PVDF nanofibers, and f-PVDF thin film, Raman spectra were recorded as shown in Figure 3b. The strong lines of α-PVDF are located at 790 cm^−1^, 890 cm^−1^, and 1430 cm^−1^. The sharp peak at 841 cm^−1^ is assigned to the symmetric CF stretching “crystallinity” band of A1 symmetry, corresponding to the β polar phase of PVDF [29]. The vibration mode of β-PVDF increases for a-PVDF nanofibers has the highest intensity compared with that of the r-PVDF nanofibers and thin films, indicating a high ratio of the polar phase in a-PVDF nanofibers. 

The FTIR spectra were further used to confirm the polar phase existing in the PVDF membranes. It was reported that the crystallographic PVDF generates absorption bands at 840 cm^−1^, 1072 cm^−1^, 1276 cm^−1^, and 1413 cm^−1^ in FTIR spectra for polar β phase, while the vibrations for the α phase generates absorption bands locate at 614 cm^−1^, 766 cm^−1^, 796 cm^−1^, 875 cm^−1^, 975 cm^−1^ and 1144 cm^−1^ [30,31]. As shown in Figure 3c, the characteristic bands for both α and β phase can be observed in the PVDF membranes. One can see that the band at 840 cm^−1^ for β phase presents high intensity, indicating the high proportion of β phase in a-PVDF nanofibers. The relative proportion of the β phase content in PVDF is usually estimated by the following equation:(1)$$F\left(\beta \right)=\frac{\chi_{\beta }}{\chi_{\beta }+\chi_{\alpha }}\times 100\%=\frac{A_{\beta }}{\frac{K_{\beta }}{K_{\alpha }}A_{\alpha }+A_{\beta }}\times 100\%$$
where *χ*_α_ and *χ*_β_ are the degrees of the PVDF α and β phases; *K*_α_ (6.1 × 10^4^ cm^2^·mol^−1^) and *K*_β_ (7.7 × 10^4^ cm^2^·mol^−1^) are the absorption coefficients at 766 and 840 cm^−1^; *A*_α_ and *A*_β_ are the absorption values at ~766 and ~840 cm^−1^, respectively, corresponding to the α and β phases shown in the FTIR spectra, respectively [10]. The β phase contents of the thick film, r-PVDF and a-PVDF nanofibers were calculated to be 53.56%, 73.31%, 74.58%, respectively. The high proportion of β phase in the highly aligned nanofibers is confirmed. It is easy to understand that during the electrospinning process, the flow of the polymer under an electric field induces the orientation of the polymer chain and polarizes the fibers. Moreover, in addition to Coulomb forces, the high-speed rotating collector can provide a mechanical stretch force to align molecular chains, which further improve the crystallinity and the formation of the polar β phase of PVDF effectively. 

The functional mechanism of piezoelectric nanogenerators can be generally described as a transient flow of electrons driven by a piezoelectric potential [32]. When an external force applies on the generator (Figure 4a), the electric dipoles align along a single direction due to the piezoelectric effect, which will generate the potential difference across the top and bottom electrodes. The generated potential drives the electrons to flow from one electrode to another through an outer circuit. When the force is released (Figure 4b), the electric dipoles reverse and the corresponding potential makes the current flow back. Thus, ac electrical signals are generated periodically under a repetitive external force. The piezoelectric energy harvesting outputs of the a-PVDF, r-PVDF and f-PVDF membranes were measured on a home-made platform in a repetitive tapping mode. The generated voltage and current were collected using a digital storage oscilloscope and the repeating frequency was set as 2.5 Hz. As shown in Figure 4c, all nanogenerators based on various forms of piezoelectric active materials can generate stable voltage and current. The generated voltages are 1 V, 8 V and 14 V for the generators assembled by thick film, r-PVDF, and highly a-PVDF membranes, respectively. The highly aligned nanofiber generator exhibits the maximum output voltage of 14 V. The generated current was also measured, as shown in Figure 4d. The generated currents were found to be in the range from 0.4 to 1.22 µA with the forms from thick film to well-aligned nanofibers, as summarized in Table 1. The high output performance of the nanogenerator based on the highly a-PVDF membranes may be due to the high proportion of ferroelectric (piezoelectric) β phase as well as the high crystallinity. Furthermore, the highly aligned structure of the PVDF nanofibers would not only increase the connection of the aligned nanofibers but also enhance the deformation of the piezoelectric membranes when subjected to external mechanical stress, and thus enhance the piezoelectric outputs. One may also note that the current in Figure 4d is much lower in the f-PVDF-based nanogenerator than that of the other two nanogenerators. According to the recent research reports, the output current of the piezoelectric nanogenerator is directly related to the Maxell’s displacement current, which can be expressed as follows [33]:(2)$$J_{z}=\frac{1}{A}\frac{\partial \left(S\sigma_{z}\right)}{\partial t}$$
where *J*_z_ is the current density, *A* is the cross sectional area of the nanogenerator, *σ**_z_* is the surface polarization charge density, and *S* represents the corresponding surface with charges. The output currents of the piezoelectric nanogenerator rely heavily on the surface polarization charges. The low output currents may due to the low content of the polar β phase in the PVDF film. Non-polar α phase can not generate surface polarization charges, resulting small value of the *σ_z_*. Furthermore, the low density of the surface polarization charges may also be attributed to the lack of the high electric field poling process. Thus the f-PVDF nanogenerators lead to low output currents.

Switching-polarity measurements are necessary to carry out checks on the output voltage that originated from the piezoelectric behavior of nanogenerators. Figure 5a,b show the output voltages of the a-PVDF generator in forward and reverse electrical connections, respectively. With the forward connection, the output voltage signals show a narrow and high positive peak and a relatively low negative peak right after the positive one. It shows the opposite signals when the generator is connected in reverse, as shown in Figure 5b. The reversible electric signals confirm that the observed voltage and current signals are generated by the piezoelectric effect due to a switching of the polarity rather than derived from environment interference [34]. One may also observe the asymmetrical positive and negative peaks in the output signals during pressing or releasing, which can be comprehended by the difference of strain rate. The pressing is applied by the external force with high strain rate and resulting in high output voltages, whereas the releasing is predominated by the resilience of the nanofibers themselves with low strain rate, resulting in low voltages [35,36].

Figure 6a,b illustrate the variation of instantaneous output voltages performance from a-PVDF, r-PVDF, and f-PVDF across various load resistances under the tapping mode of 10 N at 2.5 Hz. All nanogenerators possess increased voltage with the increase of the load resistance, which is the partial voltage on the high resistance load. Notably, the highly a-PVDF based nanogenerator shows the highest voltage among nanogenerators with different forms of PVDF, in which voltage increases from 2.5 V at 5 MΩ to 12.5 V at 500 MΩ. The maximum power density of the a-PVDF nanogenerator is calculated to be 0.267 µW/cm^2^ at 50 MΩ. 

In order to evaluate the charge storage capabilities from the as-prepared PVDF-based nanogenerators for powering electronic devices, these were demonstrated by charging a commercial capacitor using a bridge rectifier circuit. Figure 7a shows the rectified output open-circuit voltage by the a-PVDF nanogenerator. The output voltages show all-positive voltage features which peak values correspond to those of the waves before rectification. The frequency of the rectified wave has doubled compared with that of the original wave which is due to the full-wave rectification. The ac output generated by the PVDF-based nanogenerator was first converted into dc power through the rectifier, then the generated dc power was used to charge the capacitor. A 47 µF capacitor was charged to 0.5 V in 500 s by the a-PVDF generator, as shown in Figure 7b. By charging for 500 s, the 47 µF capacitor was charged to a DC voltage of 0.5 V, 0.357 V and 0.17 V by the a-PVDF, r-PVDF and film nanogenerators, respectively. The inset of Figure 7b shows the enlarged plot of accumulated voltage-time charging curves. The voltage increased continuously with a stepwise feature corresponding to the repetitive tapping and releasing. These results demonstrate that the flexible piezoelectric energy harvester can be a power supply for wearable electronic devices. 

## 4. Conclusions

In summary, we have fabricated three types of flexible PVDF piezoelectric materials of different microstructures by controlling the rotation speed of the collector during the electrospinning process. The well-aligned nanofibers possess the highest content of polar phase compared with that of the random oriented nanofibers and thick films. The high content of the piezoelectric phase in the highly aligned PVDF nanofibers resulted in superior mechanical energy conversion ability. Under an applied pressure, the a-PVDF can generate a maximum output voltage as high as 14 V and a short-circuit over 1.22 µA. These results confirm that the well-aligned piezoelectric PVDF generator has great potential in practical application for wearable electronic equipment. The proposed reliable, simple and cost-effective method without any secondary processes is a promising strategy in fabrication polymer nanofibers for flexible electronic devices. It also sheds some light on the design of the flexible piezoelectric polymer-based nanogenerators and portable/wearable personal electronics.

## Figures and Tables

**Figure 1 polymers-13-03252-f001:**
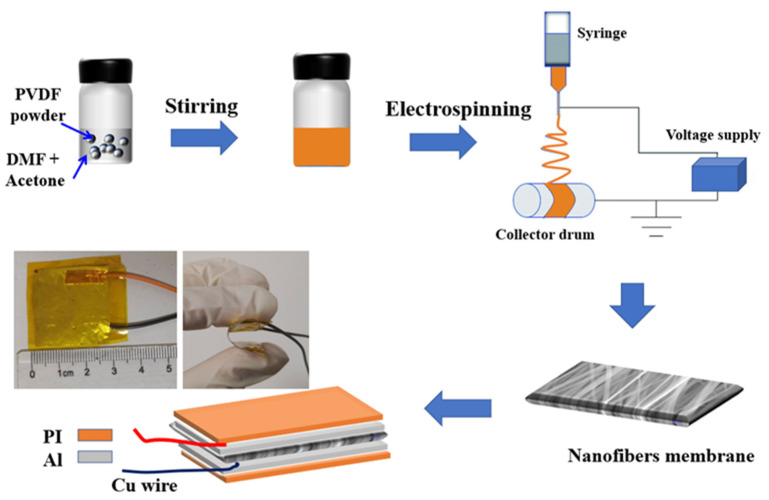
Schematic of step-wise fabrication of the nanogenerator and photographs of the flexible PVDF-based nanogenerator.

**Figure 2 polymers-13-03252-f002:**
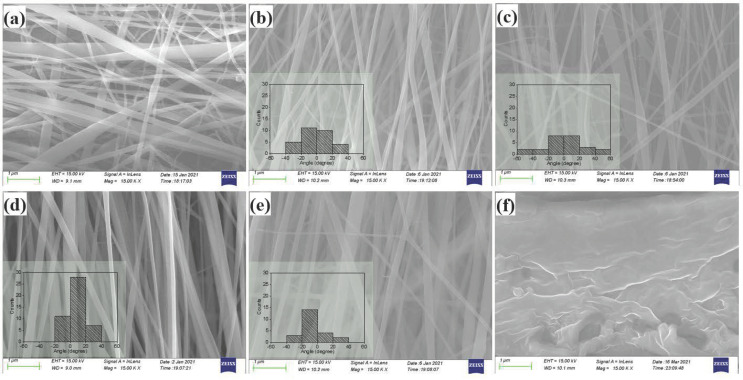
SEM images of the random oriented nanofibers (**a**) and nanofibers electrospun with a collector speed of 1000 rpm (**b**), 1500 rpm (**c**), 1700 rpm (**d**) and 2500 rpm (**e**) and cross-sectional of the PVDF film (**f**). Insets in (**b**–**e**) are orientation distribution statistics of the nanofibers.

**Figure 3 polymers-13-03252-f003:**
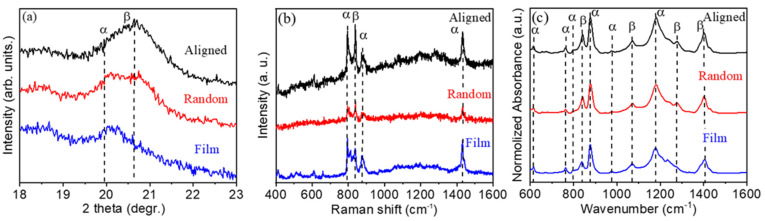
XRD pattern (**a**), Raman spectra (**b**) and FTIR spectra (**c**) of the a-PVDF, r-PVDF nanofibers and f-PVDF film, respectively.

**Figure 4 polymers-13-03252-f004:**
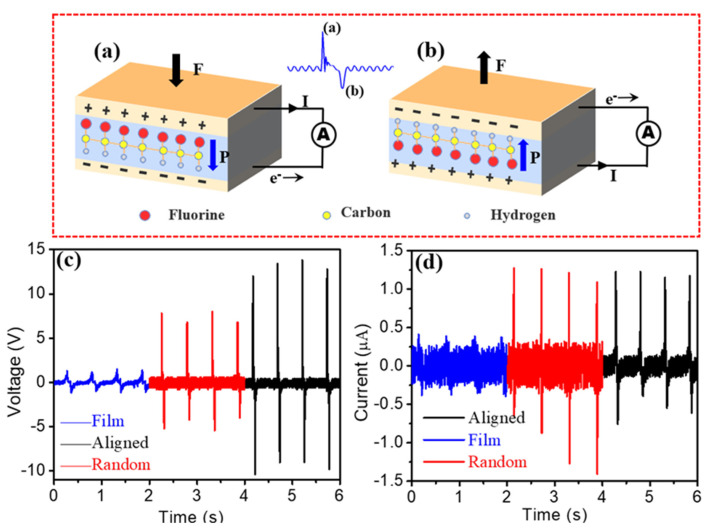
Working mechanism of the piezoelectric nanogenerator, stress applied (**a**) and released (**b**); output voltages (**c**) and short-circuit current signals (**d**) of the flexible nanogenerators assembled by thick film, r-PVDF, and highly a-PVDF membranes, respectively.

**Figure 5 polymers-13-03252-f005:**
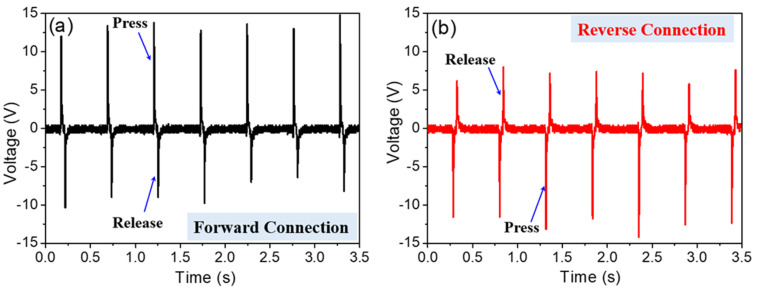
Output voltage signals from a-PVDF nanogenerator with (**a**) forward and (**b**) reverse electrical connections with measuring instrument.

**Figure 6 polymers-13-03252-f006:**
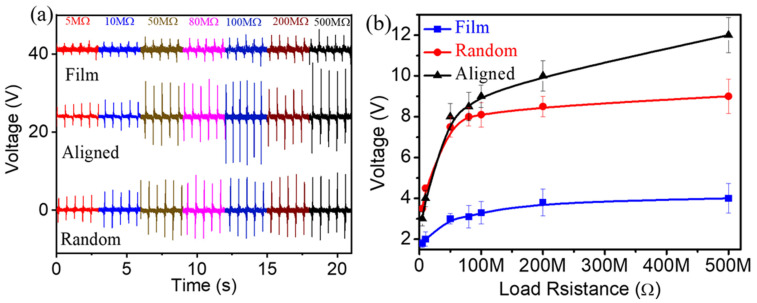
(**a**) Output voltages under various load resistances from nanogenerators assembled by a-PVDF, r-PVDF, and f-PVDF membranes, respectively. (**b**) Variation of output voltages under various external resistances.

**Figure 7 polymers-13-03252-f007:**
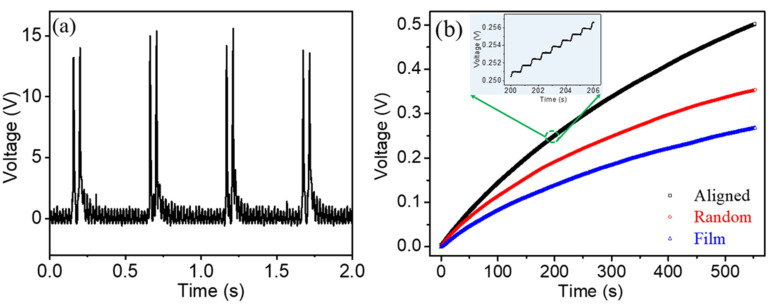
(**a**) Rectified output open-circuit voltage generated by the a-PVDF nanogenerator, (**b**) the time dependent charging curve of a 47 µF capacitor yielded from thick film, r-PVDF, and highly a-PVDF membranes nanogenerators, respectively.

**Table 1 polymers-13-03252-t001:** Sample information and performances of PVDF-based piezoelectric nanogenerators.

Sample No.	1	2	3	4	5	6
Short name	f-PVDF	r-PVDF	-	-	a-PVDF	-
Collector rotation speed (rpm)	Thin film	0	1000	1500	1700	2500
Description	Dense film	Random	Poor aligned	Poor aligned	highly aligned	Partial aligned
Voltage (V)	1	8	-	-	14	-
Current (µA)	0.40	1.20	-	-	1.22	-

## Data Availability

The data presented in this study are available on request from the corresponding author.

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
