# Peer review of "Microstructure Dependence of Output Performance in Flexible PVDF Piezoelectric Nanogenerators"

_polymers, 2021, doi:10.3390/polym13193252_

Round 1
Reviewer 1 Report
In this paper, three types of flexible polyvinylidene fluoride (PVDF) piezoelectric nanogenerators fabricated, based on well-aligned nanofibers, randomly oriented nanofibers and thick films, were investigated. The paper is interesting for the audience and can be recommended for publication after major revision.
In Introduction, more should be shown and explained what other studies about this topic already exist in this area and what results they have. For example, these papers might be a useful example:
New J. Chem., 2019,43, 284-294
ACS Appl. Electron. Mater. 2020, 2, 7, 1970-1980
J Mater Sci: Mater Electron 32, 6358-6368 (2021)
Overall, there is a lack of a good structured description of materials and methods used. This should be improved. From which manunfacturer, country of manufacturer etc. did PVDF and DMF originate? What was the exact description of these materials?
Furthermore, the electrospinning parameters are not described exactly. Here flow rate, nozzle size, at which temperature was electrospun, HR etc. should be added. It is also not clear which electrospinning machine was used, from which manufacturer, model, etc.
Furthermore, nanofibre diameter is mentioned. How was this determined and with which method and how exactly?
For a better overview and understanding, a table should provide an overview of the samples.
Also the photos of finished samples as well as a better traceable manufacturing process of samples will help audiences to understand better the process by repetition of the experiments. Here, a schematic representation of fabrication of samples should be shown and explained briefly.
It would also be useful to show schematically the working mechanism of piezoelectric nanogenerator.
The authors say that PVDF piezoelectric nanogenerators are flexible. How was the flexibility defined? Here a photo showing the flexibility of PVDF piezoelectric nanogenerators is helpful.
Conclusion part could be extended.
Author Response
We also thank the Reviewers for their time in studying our manuscript and for the valuable comments that they have provided. We provide responses that carefully address each individual comment. We have amended and improved our manuscript based on these comments and suggestions, and have also included additional editorial changes in our revised manuscript. And the response Word file is also sumitted.

Reviewer 2 Report
This paper reports a Nice and interesting research about flexible PDVF piezoeletric nanogenerators in terms of microstructure dependence for performance. I have some minor details to be addressed.
1. Motivation and impact compared with actual state of the art is not well clear.
2. Fig. 3b. Better explanation for the readers why the peaks are weak for film case.
3. How many probes or samples were tested? How it can influence the results in general?
4. How was found the optimal parameters of microstructure to get the best performance? More details in a table to check some parameters values used for such findings would be interesting.
Author Response
We also thank the Reviewers for their time in studying our manuscript and for the valuable comments that they have provided. We provide responses that carefully address each individual comment. We have amended and improved our manuscript based on these comments and suggestions, and have also included additional editorial changes in our revised manuscript. And the reponse file is also uploaded.

Reviewer 3 Report
Dear Authors
I have reviewed the article. The theme of the article is very interesting and promising in the field. Overall, the structure and content of article is acceptable for the Polymers. I am pleased to send you minor level comments. The manuscript can be accepted for publication after modification. Please consider these comments/suggestions as listed below.
- Please include one introductory line in abstract in the beginning. The rest are presented well.
- Please don’t use lumpy reference (such as [1,2]). Each reference needs to be properly addressed. Please revise and considered here this single reference ``Role of nanomaterials in the treatment of wastewater: A review. This contain same information as author stated.
- The novelty of the work must be clearly addressed and discussed, compare your research with existing research findings and highlight novelty, (compare your work with existing research findings and highlight novelty).
- The main objective of the work must be written on the more clear and more concise way at the end of introduction section.
- Please include a material section what material you use and write with chemical specifications.
- The results and discussion part seems fine. I don’t have any comment in this section.
- However, regarding the replications, authors confirmed that replications of experiment were carried out. However, these results are not shown in the manuscript, how many replicated were carried out by experiment? Results seem to be related to a unique experiment. Please, clarify whether the results of this document are from a single experiment or from an average resulting from replications. If replicated were carried out, the use of average data is required as well as the standard deviation in the results and figures shown throughout the manuscript. In case of showing only one replicate explain why only one is shown and include the standard deviations.
- Conclusion should be rewrite completely.
- Conclusion section is missing some perspective related to the future research work, quantify main research findings, highlight relevance of the work.
- Please carefully check the moderate level English language there are few grammatical errors throughout the text. These are suggestions for improving the text, not criticisms of the work's quality, which is excellent.
Author Response
We also thank the Reviewers for their time in studying our manuscript and for the valuable comments that they have provided. We provide responses that carefully address each individual comment. We have amended and improved our manuscript based on these comments and suggestions, and have also included additional editorial changes in our revised manuscript. And the response file is also uploaded.

Round 2
Reviewer 1 Report
How many nanofibers were counted to determine fiber diameter?
Do you have more specific information about PVDF and DMF?
Table 2. Why there are no short names at sample 3,4 and 6. If they do not contain relevant information, they can be deleted.
Author Response
We thank the Reviewers for their time in studying our manuscript and for the valuable comments that they have provided. We provide responses that carefully address each individual comment. We have amended and improved our manuscript based on these comments and suggestions in our revised manuscript. All the revisions have been made with edit track in the revised manuscript. The response file was also uploaded.

This manuscript is a resubmission of an earlier submission. The following is a list of the peer review reports and author responses from that submission.